# Apple Pomace and Performance, Intestinal Morphology and Microbiota of Weaned Piglets—A Weaning Strategy for Gut Health?

**DOI:** 10.3390/microorganisms9030572

**Published:** 2021-03-10

**Authors:** Sandrine Dufourny, Nadine Antoine, Elena Pitchugina, Véronique Delcenserie, Stéphane Godbout, Caroline Douny, Marie-Louise Scippo, Eric Froidmont, Pierre Rondia, José Wavreille, Nadia Everaert

**Affiliations:** 1Walloon Agricultural Research Centre, 5030 Gembloux, Belgium; e.pitchugina@cra.wallonie.be (E.P.); e.froidmont@cra.wallonie.be (E.F.); p.rondia@cra.wallonie.be (P.R.); J.wavreille@cra.wallonie.be (J.W.); 2Fundamental and Applied Research for Animals & Health (FARAH), University of Liège, 4000 Liège, Belgium; Nadine.Antoine@uliege.be (N.A.); veronique.delcenserie@ulg.ac.be (V.D.); cdouny@uliege.be (C.D.); mlscippo@uliege.be (M.-L.S.); 3Research and Development Institute for the Agri-Environment, Quebec, QC G1P 3W8, Canada; stephane.godbout@irda.qc.ca; 4Teaching and Research Centre (TERRA), University of Liège, 5030 Gembloux, Belgium; nadia.everaert@uliege.be

**Keywords:** apple pomace, piglet, weaning, microbiota, biomolecules, gut health

## Abstract

Apple pomace (AP) is known to be rich in biomolecules beneficial for health and it may advantageously be used to overcome the critical step of piglets’ weaning. The study aimed to determine the effect of two levels of incorporation of AP on the performance, intestinal morphology, and microbiota of weaned piglets and investigate this feed ingredient as a weaning strategy. An experiment was performed with 42 piglets from weaning (28 days old) over a five-week period, including three iso-energetic and iso-nitrogenous diets (0%, 2%, and 4% dried AP diets) with seven pen-repetitions per diet (two pigs per pen). AP diets were beneficial for the average daily gain calculated on week 3 (*p* = 0.038) and some parameters of the intestinal architecture on the 35 post-weaning day. The 4% AP diet was beneficial for the feed conversion ratio (*p* = 0.002) and the energetic feed efficiency (*p* = 0.004) on the 35 post-weaning day. AP tended to influence the consistency of feces (softer to liquid, *p* = 0.096) and increased the counts of excreted pathogens (*p* = 0.072). Four percent AP influenced the richness of the microbiota and the bacteria profile as observed for the phylum Bacteroidetes or the class Clostridia. The 4% AP diet appeared as an interesting weaning strategy that should be evaluated in a large cohort.

## 1. Introduction

Apple is a fruit rich in phytochemicals linked with good health indicators for humans (e.g., decreased risk of cancer or cardiovascular disease, positively associated with general pulmonary health); in particular, its content in both dietary fiber and phenolic compounds may partially contribute to this beneficial effect [1]. Similarly, the product derived from apple juice extraction process, apple pomace (AP) is known to contain biomolecules beneficial for health such as phenolic and terpenic compounds [2,3]. AP can represent a raw material of great interest for different applications such as the production of lactic acid [4], as well as being a source of dietary fiber or polyphenolic components [5,6], and as a possible functional food for the agri-food sector [7]. AP seems to have positive health effects in rats when used as a feed ingredient. Indeed, it improved the antioxidant status of the animals—by increasing the activity of superoxide dismutase in a hemolysate of erythrocytes and by increasing the antioxidant capacity of the lipid fraction of serum. It also reduced the blood glucose level of the rats and increased the fermentation process in the distal part of their gastrointestinal tract [8]. Incorporated at 3.5% in a weaner diet for piglet, beneficial effects were observed on the gut morphology of the animals [9] or on other parameters such as intestinal bacteria, blood parameters, or gene expression of immunological markers [10].

Although beneficial effects may thus be expected, it must be kept in mind that the composition of AP can largely vary in space and time. For example, the apple cultivars used for the juice extraction can influence the physico-chemical properties and antioxidant activities of the residual AP [11,12]. The use of enzymes for extraction of juice [13] or the blanching and drying process employed on AP [14] have an impact on its composition. So the potential variability in the composition of the matrix encourages continuing research on this healthy raw material for its introduction in the feed chain at a large scale and shows the importance, for feed producers, of properly characterizing the matrix before using it.

Weaning is a critical step in pig production. Indeed, weaning is generally an abrupt and precocious event—3 to 5 weeks after parturition in the current production—and leads to multiple external factors of stress included change in the diet from a 20% dry matter milk feed to an 80–90% dry matter plant-based diet, change in physical and social environments including the maternal separation and the disruption of the established social group [15,16]. In a few hours, the piglet has to adapt to a novel environment including unfamiliar feeding and drinking equipment, unfamiliar piglets, with no more stimulus from the sow to start feeding, and potential lack of thermal comfort [15]. All these stressors induce a reduction of piglets’ feed intake and multiple biological stresses related to the intestinal structure and functions, to the nervous and hormonal pathways and to the immune system that may have an impact on short and long-term performance and health status of the piglet [17,18,19].

In a few days after weaning, the young intestinal structure has to adapt to the new diet. Modifications of the intestinal architecture and functions relate to the decrease of villi height, the increase of crypts depth and intestinal cell mitosis, the reduction of brush border enzyme activity and the reduction of the absorptive capacity of nutrients and electrolytes [15]. In addition to these physiological disturbances, impacts on the gut barrier function, the piglets’ immunity and the microbiota are also described. It induces an inflammatory response of the intestinal mucosa and makes the animal sensitive to antigens, toxins, and translocation of bacteria [17,20,21]. As a consequence of all these disturbances, weaning generates post-weaning symptoms in piglets, including post-weaning diarrhea often induced by the enterotoxigenic *Escherichia coli* [22].

Weaning requires nutritional management [23] in order to prevent or counteract the negative effects on growth performance and intestinal disorders [24] and to ensure lifelong gut health for pigs [25]. AP at an incorporation dose of 3.5% is a promising matrix [9,10]. However, technical and economic constraints linked with the production of dried AP, may wonder whether a lower dose of AP also results in beneficial effects in weaned piglets. Consequently, the aim of the study was to evaluate the effects of dried AP—incorporated at two levels into a post-weaning diet (a positive control level set at 4% and a lower intermediate level set at 2%)—on growth performance, on intestinal morphology and on the microbiota—of the feces during the post-weaning as well as that of the caecum at the end of the post-weaning period—and globally discuss about AP as a weaning strategy. AP diets, particularly the higher dose, showed positive effects on some performance indicators explained partially by the intestinal architecture. The feces of the 4%AP piglets tended to be softer, to count more excreted pathogens and to have an enriched microbiota. In the caecum on the 35 post-weaning day, a differential effect on the microbiota was more observed between 2% AP and 0% AP. The microbiota of 4% AP was similar to 0% AP. Taking into account globally the results of the experiment, the 4% AP diet appeared as an interesting weaning strategy that merits to be evaluated in a large cohort to evaluate finely the risk of dysbiosis due to the excretion of pathogens in the beginning of the post-weaning period.

## 2. Materials and Methods

### 2.1. Experimental Design

#### 2.1.1. Animals, Diets and Housing

The intervention on piglets was approved by the ethical committee of the University of Liège (ULiège, Liège, Belgium)—file n°1823. The intervention was in compliance with European (Directive 2010/63/EU) and Belgian (Royal Decree of the 29 May 2013) regulations governing the protection of animals used for scientific purposes.

Forty-two piglets [Pietrain x Landrace] from the Walloon Agricultural Research Centre (CRA-W, Gembloux, Belgium)—21 females and 21 castrated males, free of antibiotics—were weaned at 28 days of age and transported to the Animal Production Centre in Gembloux (CEPA-University of Liège, Gembloux, Belgium) for 35 days of post-weaning rearing.

During the lactation period, piglets received a creep feed (meal dry feed) for 10 days before weaning. The creep feed was composed of 50% milk powder and 50% of the control diet (SCAR, Herve, Belgium). During the post-weaning period, piglets received one of the three diets (meal dry feed) formulated for the experiment (control diet with no AP—0% AP-, experimental diet containing 2% dried AP—2% AP-, experimental diet containing 4% dried AP—4% AP-; Table 1; SCAR, Herve, Belgium). Diets were formulated to be iso-energetic (9.6 MJ EN/kg) and iso-nitrogenous (17.5% CP), having in mind the recommendations of the NRC. To formulate the experimental AP diets, a portion of wheat was removed from the 0% AP diet that was substituted by AP and—in a lesser extent—by soybean meal and soya oil.

For the post-weaning housing, piglets were assigned in pair (1 female and 1 male) to seven boxes (1.5 m^2^, polymer grating) for each of the three diets following a randomized complete block design. The boxes of the piglets were gathered in 7 blocks (including each the 3 diets), taking into account a potential thermic gradient in the housing.

Piglets were fed ad libitum and the enrichment of the boxes was done with steel chains. All boxes were washed with water every day after scoring the piglet’s fecal consistency.

#### 2.1.2. Sampling of Feces

A sample of feces was collected directly from the rectum of each male two times, i.e., on day 8 and day 28, during the post-weaning period. A part of the samples of day 8 was directly used fresh to detect excreted pathogens. All the samples (day 8 and day 28) were stored at −80 °C, until DNA extraction.

#### 2.1.3. Sampling of Intestinal Tissues and Contents

On the last day of the experimental period—day 35—the male piglets were euthanized by isoflurane inhalation followed by bleeding. Their gastrointestinal tract was removed and each intestinal segment was isolated. The chyme and the mucosal layer of the ileum (collected from 1 m to 50 cm of the ileo-cecal junction), caecum and proximal colon (collected from 30 cm after the ileo-cecal junction) were immediately snap-frozen in liquid nitrogen and maintained at −80 °C until DNA extraction. Five cm of tissue samples from the duodenum (from 25 cm after the stomach), jejunum (from 2 m after the stomach) and ileum were collected, rinsed with a saline solution and fixed in 10% formol until morphological analyses.

### 2.2. Experimental Measurements

#### 2.2.1. Chemical Analyses

AP was chemically analyzed for the mono-, and disaccharides, fiber, and polyphenols profile by the Biomass and Green Technologies laboratory (University of Liège, Gembloux, Belgium) using gas chromatography (GC) after derivatization [26] and high-performance anion exchange chromatography (HPAEC-PAD) [27]. Rhamnose, arabinose, xylose, mannose, galactose, and glucose were determined after aqueous extraction by GC including derivatization step. Saccharose and fructose were determined in the aqueous extract by HPAEC-PAD. Arabino-, xylo- and galacto-oligosaccharides were determined by GC after acid hydrolysis and derivatization. The solvent used was dichloromethane. Glucuronic acid, galacturonic acid and constituting sugars were determined by HPAEC-PAD and GC after hydrolysis. Total polyphenols were determined, after extraction of the phenolic compounds with a polar solvent, by spectrophotometry after oxidation of the phenolic compounds by the Folin–Ciocalteu reagent [28].

Diets were chemically analyzed by CRA-W (Gembloux, Belgium) for their content in humidity, crude protein, crude ash, reducing sugars, total sugars, starch, cellulose and crude fat following Commission Regulation (EC) No 152/2009 of 27 January 2009 laying down the methods of sampling and analysis for the official control of feed. The NDF, ADF, ADL content were analyzed following Standard NFV18-122 August 1997 for animal feedstuffs—Determination of sequential cell-wall—Method by treatment with neutral and acid detergent and sulfuric acid.

AP and diets were chemically analyzed by Upscience (Saint-Nolff, France) for their insoluble high molar weight dietary fiber (HMWDF), soluble HMWDF and soluble low molar weight dietary fiber (LMWDF) following AOAC 991.43 method.

#### 2.2.2. Zootechnical Performance

Piglets were weighed (Giropes G1308, Pesage Warnier, Hannut, Belgium) on a weekly basis, with the initial weight being the weaning weight. The average daily gain (ADG) was calculated per box, on a weekly and cumulative basis (from initial weighting to specific date of weighting), with the final ADG being those calculated using the initial weight and final weight of piglets. The total feed intake (TFI) and the feed conversion ratio (FCR) were calculated at the end of the experiment per box. The energetic feed efficiency (EFE) was calculated as the ratio between the total net energy of the diet ingested per box and the total weight gain of the piglets of the box. The net energy of the diet was calculated with EvaPig^®^ software (v 1.3.1.7, INRA–AFZ–Ajinomoto Eurolysine S.A.S., Paris, France) using the results of the feed chemical analyses.

#### 2.2.3. Scoring of Piglet’s Fecal Consistency

The consistency of the piglet’s feces was visually evaluated on a daily basis per box from the second day of weaning until the 15 post-weaning day. Two scores were attributed per box (minimum and maximum scores). The rating scale included 6 ranks [29] from score 0 to score 5. The score 0 was an absence of feces. The score 1 was multiple free pellets of feces. The score 2 was aggregated pellets shaping the feces. The score 3 was firm feces, shaped as a cylinder). The score 4 was soft feces, not shaped as a cylinder. The score 5 was liquid feces.

#### 2.2.4. Excreted Pathogens

The presence/abundance of pathogens that are known to be involved in the diarrhea of piglets was investigated on the 8 post-weaning day through the use of the Rainbow Piglet ScoursTM Bio K 351 (Bio-X Diagnostics S.A., Rochefort, Belgium)—detecting Rotavirus, *E. coli* F4, F5, F18, F41 attachment factors, *Clostridium difficile*, *Clostridium perfringens* and Cryptosporidium—following the manufacturer’s instructions.

#### 2.2.5. Intestinal Morphology

The morphological measurements of the intestinal segments were determined following a protocol previously described [30]. The segments, fixed in 10% formol, were dehydrated and embedded in paraffin wax. For each intestinal segment, four sections of 5 μm were obtained and stained with Alcian blue for mucous detection following routine methods. The sections obtained were scanned, digitized using an imaging system for virtual microscopy (Dotslide, Olympus, Belgium) and analyzed with a Java image morphometric processing program (Image J software, National Institute of Health, Bethesda, MD, USA). Villus length and crypt depth were measured in order to obtain a total of 20 measurements of crypt and villus per section (80 per pig and per intestinal segment).

#### 2.2.6. Short Chain Fatty Acids (SCFA)

The SCFA analyzed were acetic (C2), propionic (C3), isobutyric (iC4), butyric (C4), isovaleric (iC5), valeric (C5) and hexanoic acids (C6). The SCFA content of the feces and chyme from the caecum was measured by SPME-GC-MS following a protocol previously described [31]. Between 20 and 25 mg of samples were introduced into a 20 mL glass vial. Forty µL of internal standard (2-methylvaleric acid) at a concentration of 0.2 mg/mL, 15 µL of 0.9 M sulfuric acid, and 920 µL of water were then added. The mixture was vortexed and placed on the autosampler of the SPME-GC-MS system until analysis. SCFA were extracted with a SPME fiber, separated on a Focus GC gas chromatograph (Thermo Fisher Scientific, Waltham, MA, USA) using a Supelcowax-10 column (30 m × 0.25 mm, 0.2 µm) (Supelco, Bellefonte, PA, USA) and analyzed with an ion trap PolarisQ mass spectrometer (Thermo Fisher Scientific, Waltham, MA, USA). The agitation temperature was set at 60 °C and the extraction time at 20 min.

#### 2.2.7. Composition and Richness of the Microbiota

The composition and richness of the microbiota were analyzed on all fecal and cecal samples. DNA extraction and sequencing of all the samples were performed by DNA Vision (Gosselies, Belgium) following their internal quality SOP. DNA was extracted from frozen pellets with the DNeasy Blood & Tissue kit according to the instructions of the manufacturer Qiagen (Qiagen Benelux B.V., Venlo, The Netherlands). DNA was quantified and qualitatively assessed on a NanoDrop 2000 from Thermo Scientific™ and by PicoGreenVICTOR X3 (PerkinElmer) using the Quant-it PicoGreen dsDNA Assay kit from Invitrogen. The 16S targeted region V3-V4 was amplified by PCR, purified and tagged. Libraries were indexed using the NEXTERA XT Index kit V2 from Illumina. The high throughput sequencing was carried out on Illumina Miseq in paired-end sequencing (2 × 250 bp) by targeting an average of 10,000 reads per sample. Finally, the bioinformatic analysis was executed with the QIIME (Quantitative Insights Into Microbial Ecology) software, version 1.9.0 with “Greengenes 13_8” as database and recommended parameters to use QIIME scripts. The OTU (Operational Taxonomic Unit) table was generated based on a 97% sequence similarity of the sequencing reads to cluster OTUs. Only samples presenting more than 5000 reads were used for taxonomic analysis. Similarly, samples with the same normalized number of reads were used for the beta diversity (OTU) analysis for which the results were expressed in relative abundance—a percentage of the total bacteria. Beta diversity (group comparisons)—representing comparison of microbial communities based on their composition—generated a matrix where dissimilarities/distances were calculated between every pair of group samples and that was visualized with Principal Coordinates Analysis. The group comparison analysis was performed for the Weighted Unifrac distance—based on the abundance of observed organisms.

The parameters of alpha-diversity that were analyzed are Chao1, observed operational taxonomic unit, phylogenetic diversity whole tree and Shannon index.

### 2.3. Statistical Analyses

#### 2.3.1. Parametric Tests

The conditions for applying the analysis of variance on the quantitative data of the experiment have been verified. The assumptions of the data normality and the equality of variances according to the treatment were confirmed.

Zootechnical performance measured at the end of the post-weaning phase, calculated as mean values for the global post-weaning period (final weight, final ADG, TFI, FCR, and EFE) and intestinal morphology measurements were analyzed using a two-way analysis of variance with “diet” as fixed factor and “block” as random factor (GLM procedure, SAS v9.4, SAS Inst. Inc., Cary, NC, USA). The Student–Newman and Keuls test was used to structure the averages. SCFA data from chyme samples were analyzed following the same procedure using Minitab software (Minitab 18, Minitab Inc., State College, PA, USA).

Zootechnical performance measured during the post-weaning period (ADG calculated per box, on a weekly and cumulative basis) and SCFA data from fecal samples were analyzed as repeated measures following a split-plot analysis of variance [32] with “diet” as fixed factor, “piglet” as random factor and “week” as fixed factor (Minitab 18, Minitab Inc., State College, PA, USA). Moreover, as a complement of information, analyses of variance were also performed separately on the different dates of the weekly ADG.

A *p*-value lower or equal to 0.05 was considered statistically significant. A *p*-value between 0.05 and 0.1, or equal to 0.1, was considered a trend. Otherwise, a *p*-value higher than 0.1 was considered not significant (ns).

#### 2.3.2. Non-Parametric Tests

The qualitative data (pathogens in feces and fecal consistency scores) were analyzed using non-parametric tests for which categories were defined in order to comply with the test application conditions.

The qualitative data obtained from the detection of pathogens by Rainbow kit were analyzed using a non-parametric chi-square test of association (Minitab 18, Minitab Inc., State College, PA, USA) ranking the diet factor into two categories (no AP in the diet and AP in the diet) and the results into two categories (negative: no pathogen detected or <10^6^ CFU/g; positive: pathogen detected or >10^6^ CFU/g).

The qualitative data obtained from the piglet’s fecal consistency were analyzed using a non-parametric Cochran test (Minitab 18, Minitab Inc., State College, PA, USA), ranking the diet factor into two categories (no AP in the diet and AP in the diet) and the results of the maximum scores into two categories (category A included the scores 0 to 3; category B included the scores 4 and 5 that was considered as more diarrheic).

The statistical analysis for the alpha-diversity data of the microbiota was based on a non-parametric t-test (Monte Carlo permutations to calculate *p*-value) comparing groups of samples two by two. The analysis for the beta-diversity data (OTU) was done at different levels of the taxonomy classification to detect differences in read abundances between groups of samples. The non-parametric Kruskal–Wallis test was used for this purpose (KW *p*-value) and it was subsequently adjusted using the Benjamini–Hochberg false discovery rate procedure for multiple comparisons (FDR *p*-value). The analysis for the beta-diversity data (group comparison, weighted Unifrac distance) was done through the Adonis statistical test.

A *p*-value lower or equal to 0.05 was considered statistically significant. A *p*-value between 0.05 and 0.1, or equal to 0.1, was considered a trend. Otherwise, a *p*-value higher than 0.1 was considered not significant (ns).

## 3. Results

### 3.1. Feed Chemical Analyses

Results of the chemical analyses are given in Table 2 for the AP and in Table 3 for the diets.

### 3.2. Zootechnical Performance

Piglets from two boxes were excluded from the experimental set-up due to health issues in the first days of the experiment. The first piglet—0% AP box—was excluded due to the infection of a hoof of the female piglet. The second piglet—4% AP box was excluded due to a paw problem of the male piglet.

For the performance at the end of the post-weaning period, final weight, final ADG and TFI were not statistically different between diets (Table 4). FCR, as well as EFE of piglets that had received the 4% AP diet were significantly lower compared to piglets of the 0% AP and 2% AP diets.

For the weekly ADG measured during the post-weaning period, the statistical analyses using repeated measures did not differ between diets. However, analyses of variance performed on ADG week by week (Table 5) revealed that ADG in week 3 was higher for 2% AP and 4% AP diet than that of 0% AP diet. ADG in week 2 and week 4 showed a trend to be different.

### 3.3. Scoring of Piglet’s Fecal Consistency and Excreted Pathogens

The results obtained from the observation of the feces (maximum score in each box) during the first two weeks of the post-weaning period showed a trend (*p* = 0.096) for more softer to liquid feces with AP diets than with the 0% AP diet (Table 6, left side). The results obtained through the use of the Rainbow kits showed a trend for more pathogens with diets containing AP than with the 0% AP diet (Table 6, right side, *p* = 0.072).

### 3.4. Intestinal Morphology

The results of the morphological measurements performed on the three small intestinal segments showed a significant effect of the diet on the duodenal villus length and on the ileal ratio villus length/crypts depth (Table 7). Piglets receiving the 4% AP diet presented higher duodenal villus length than piglets from the 0% AP diet; with intermediate values for the 2% AP piglets. In the same way, piglets receiving the 4% AP diet had a higher ileal ratio villus length/crypt depth than piglets with the 0% AP diet (with the difference between 4% AP and 0% AP mainly due to the higher villus length of 4% AP in mean value). No effect of the diet was observed on jejunum measurements although numerical differences between the mean values in jejunum were similar to those observed in the duodenum.

### 3.5. Short Chain Fatty Acids (SCFA)

No SCFA differences between groups were observed in the samples of feces on the 8 and 28 post-weaning days (calculated as g/kg feces or as mmol/kg feces) and in the caecum samples (calculated as g/kg cecal content or as mmol/kg cecal content) (data not shown).

### 3.6. Richness and Composition of the Microbiota

#### 3.6.1. Fecal Microbiota Composition on Day 8 Post-Weaning

The Shannon index tended (*p* = 0.094) to be higher in the feces of 4% AP piglets (5.7) than in the feces of 0% AP piglets (5.0), while the Shannon index of the 2% AP piglet’s feces was 5.4. Other indices (Chao1, observed operational taxonomic unit, phylogenetic diversity whole tree) were not different. The weighted Unifrac distance tended to be different between groups (*p* = 0.052, Appendix B).

At the phylum level of classification (Figure 1), for the 0%AP and 2% AP piglets, the phyla showing the higher relative abundances—by decreasing order—were Firmicutes, Actinobacteria and Bacteroidetes (containing Bacteroidia as unique class in all the samples of the study). For the 4% AP piglets, it was Firmicutes, then Bacteroidetes that was more abundant than Actinobacteria. For all piglets, the phylum Proteobacteria was scarce (0.0% for 0% AP, 0.1% for 2% AP, and 0.5% for 4% AP).

The class Clostridia was twice as abundant in comparison to Bacilli in apple pomace diets while they were equal in the diet without AP (Figure 1).

Coriobacteriaceae, Lactobacillaceae and Enterobacteriaceae were bacterial families influenced by the diet as well as *Dorea, Slackia*, *Ruminococcus*, and *Catenibacterium* when considering the genus level of the classification (Appendix C and Appendix A).

#### 3.6.2. Fecal Microbiota Composition on Day 28 Post-Weaning

The indexes describing the α-diversity did not differ between the diets. The weighted Unifrac distance tended to be different between groups (*p* = 0.069, Appendix B).

As observed on the 8 post-weaning day, the phylum Firmicutes showed the highest relative abundance in the fecal microbiota of piglets on the 28 post-weaning day (Figure 2). It was then followed for the three diets by the phylum Bacteroidetes and then Actinobacteria. The relative abundance of Proteobacteria was below 1% (0.2% for 0% AP, 0.1% for 2% AP, and 0.2% for 4% AP; *p* = not significant).

The class Clostridia was the first class of bacteria (>50% relative abundance) for AP diets and it became also the first class for the 0% AP diet at this timepoint (Figure 2).

Amongst the more abundant families—cited by decreasing order—were Lactobacillaceae, Lachnospiraceae, and Ruminococcaceae for the 0% AP and 2% AP piglets. It was Lactobacillaceae, Ruminococcaceae, and Lachnospiraceae for the 4% AP diet (Table 8). Between these three families, Ruminococcaceae showed a significantly higher relative abundance for the 4% AP diet than for the 2% AP diet (*p* = 0.047) and a trend to be more abundant than for the 0% AP diet (*p* = 0.078). Clostridiaceae showed also a significantly higher relative abundance for the 4% AP diet than for the 2% AP diet (*p* = 0.028); 0% AP was intermediate and not statistically different from 2% AP and 4% AP. Veillonellaceae showed the lowest abundance for the 4% AP diet than for the 0% AP and 2% AP diet (*p* = 0.009).

At the genus/species levels of the classification (Table 8, Appendix A), two bacteria from the Clostridiaceae family were significantly more abundant in the feces of the 4% AP piglets (SMB 53_undefined species and Clostridiaceae_undefined genus) compared to the 2% AP piglets, with intermediate values for the 0% AP group. In contrast, *Blautia*_other sp., *Dorea*_other sp.—both from Lachnospiraceae family—and undefined species of *Dialister*, *Megasphaera*, and *Mitsuokella*—these three bacteria from the Veillonellaceae family—were more abundant in the 0% AP group compared to the 4% AP; results for the 2% AP diet was statistically not different from 0% AP or 4% AP following the bacteria as seen in Table 8.

#### 3.6.3. Cecal Microbiota Composition on Day 35 Post-Weaning

The indices describing the α-diversity of the cecal microbiota of the chyme (Chao 1, number of operational taxonomic unit and phylogenetic diversity) revealed an increased richness for the 4% AP piglets compared to the 2% AP piglets (*p*-value significant for the three parameters), the indices for 0% AP were intermediate (Table 9). The weighted Unifrac distance was not different between groups (*p* = ns, Appendix B).The profile of the microbiota present in the chyme of the caecum (data not shown) was similar between diets except for *Coprococcus* (“undefined species”) that was more abundant in the 0% AP piglets (1.3%, *p* = 0.014) than in the 2% AP piglets (0.6%), while the abundance for the 4% AP piglets was intermediate (1.1%). A trend was also observed for Muribaculaceae (0.1% for 0% AP, 0.1% for 2% AP, 0.2% for 4% AP; *p* = 0.078) and *Lachnospira* (“undefined species”; 0.1% for 0% AP, 0.4% for 2% AP, 0.5% for 4% AP; *p* = 0.069).

The indices describing the α-diversity of the cecal microbiota of the mucosa were less influenced by the diet than chyme samples. Chao 1 was the only index of the α-diversity of the microbiota to be statistically higher for 4% AP piglets compared to 2% AP piglets, while 0% AP had an intermediate value (Table 9). The number of operational taxonomic units tended to be more substantial for 4% AP piglets. The weighted Unifrac distance tended to be different between groups (*p* = 0.054, Appendix B). From a bacterial point of view (Table 10), more differences were visible between diets compared to the chyme results. Firmicutes were more abundant for 2% AP piglets (*p* = 0.045) than for 0% AP and 4% AP piglets; Bacteroidetes were more abundant for 0% AP and 4% AP piglets (*p* = 0.049) than for 2% AP piglets. The third dominant phylum—Proteobacteria—tended (*p* = 0.053) to be more abundant in 0% AP piglets (5.6%) and less abundant in 2% AP piglets (1.3%). Statistical differences or trends were observed for families Muribaculaceae, Pasteurellaceae, Peptostreptococcaceae and Prevotellaceae as well as for species *Acidaminococcus* “undefined species”, *Actinobacillus* “other species”, *Campylobacter* (“undefined species”), *Coprococcus* “undefined species”, *Megamonas* “undefined species”, *Mitsuokella multacida*, *Oscillospira* “undefined species” and *Prevotella* “undefined species”.

## 4. Discussion

### 4.1. Zootechnical Performance and Intestinal Morphology

During the post-weaning period, AP improved the ADG of piglets on week 3. At the end of the post-weaning period, only piglets that had received the 4% AP diet showed a lower FCR and EFE compared to 0% AP piglets, which may partially be explained by the improved intestinal morphology, i.e., a higher villus length in the duodenum and a higher VL/CD in the ileum compared to the 0% AP piglets. Indeed, the villus length/crypt depth ratio is a good indicator for estimating the likely digestive capacity of the small intestine [33]. These observations are in line with those of Sehm and colleagues [9] who showed a beneficial effect of 3.5% of dry AP on the villus height at certain time points of the post-weaning period. It could be hypothesized that this improved small intestinal morphology may be explained by the higher level of reducing sugars in the AP diets (mainly in the 4% AP diet), particularly due to fructose from AP. Indeed, AP is rich in fructose and it was demonstrated—in humans—that fructose is primarily metabolized in the small intestine [34], where it is an interesting source of energy for the enterocytes. As the zootechnical performance at the end of post-weaning for the 2% AP piglets were similar to those of 0% AP piglets and as the intestinal morphology results were intermediate and statistically not different from those of 0% AP and 4% AP piglets, reducing the level of incorporation of AP in the diet seems not appropriate concerning zootechnical performance mainly due to a stunted growth at the beginning of the post-weaning period.

### 4.2. Scoring of Piglet’s Fecal Consistency and Excreted Pathogens

On the eighth post-weaning day, more pathogens tended to be detected in the feces of piglets receiving AP (through the Rainbow kit), together with the presence of numerically softer/liquid feces during the first two post-weaning weeks. It is known that healthy individuals can live in equilibrium with pathogens without clinical symptoms as long as there is equilibrium between the host and its entire microbiota [35]. On the one hand, the softer and liquid feces observed could be due to the pathogens [33], which can represent a threat for the piglets’ health, leading to a potential use of antibiotics. On the other hand, we cannot exclude the hypothesis that the softer feces can be due to the fiber composition of the AP diets. Indeed, the water holding capacity of some fiber [36] and the possible abrasive effect of large/coarse fiber [37] may affect the intestinal mucosa and the consistency of the feces. This invites further unravelling the reasons why softer feces are observed with AP to confirm or refute the risk of an increasing use of antibiotics with AP, which can compromise the use of this ingredient as a weaning strategy. Under the good sanitary conditions offered during the experiment, pathogens did not appear as a threat and were not the cause of poor performance. However, it would be of great interest to test AP in large cohort to assess the risk of dysbiosis with a 4% AP diet and confirm or refute the ability of AP to maintain homeostasis or the absence of clinical symptoms.

### 4.3. Microbiota and SCFA in Feces and Cecum

Although some bacterial indicators on the 8 post-weaning day may appear less favourable with AP (e.g., Enterobacteriaceae more abundant in 2% AP feces than 0% AP feces or Lactobacillaceae less abundant with AP), several indicators point to a positive effect of a 4% AP addition in the post-weaning diet.

Firstly, the richness of the microbiota in the feces seems better for the 4% AP diet for this critical period (eighth post-weaning day), which is an important factor for health [38], increasing ecological stability and resilience after a stress-related disturbance [39]. Furthermore, this higher α-diversity for the 4% AP group was clearly observed in the chyme and on the mucosa of the caecum on the 35 post-weaning day.

Secondly, the second most abundant phylum in the feces for the 4% AP piglets on the eighth post-weaning day was Bacteroidetes which can also be considered as positive, seen their potential for microbial enrichment [40] or metabolism modulation for human health [41]. In line with this high relative abundance of Bacteroidetes in feces of piglets from the 4% AP diet on the 8 post-weaning day—reflecting the fermentation process at the end of the large intestine—the Bacteroidetes were also highly present in the cecal mucosa on the 35 post-weaning day—as a result of the fermentation process at the beginning of the large intestine. It should however be noted that in the cecal mucosa on the 35 post-weaning day, the relative abundance of the Bacteroidetes were also high in the 0% group, as compared to the 2% AP group, but the Bacteroidetes were the third phylum in feces of 0% AP and 2% AP piglets, on the 8 post-weaning day. However, caution is required when interpreting the results for Bacteroidetes because this phylum also includes bacteria that can become problematic for health, as seen with *Prevotella copri*, acting in a beneficial or detrimental manner depending on the context [42,43].

Thirdly, the higher abundance of Clostridia in the feces of the AP group on the eighth post-weaning day further supports the beneficial effect of AP, seen their importance for the maintenance of immune and gut homeostasis [44,45].

Fourthly, the higher relative abundance of the Ruminococcaceae in the feces of 4% AP piglets compared to those of 0% AP and 2% AP piglets on the 28 post-weaning day, is in line with a possible improved resistance/tolerance to pathogens [46]. In the cecal mucosa again, both the 0% and 4% AP groups had the highest relative abundance of Ruminococcaceae compared to the 2% AP group.

Lastly, although only a trend, the higher relative abundance of Proteobacteria in the mucosa of the caecum of 0% AP piglets on the 35 post-weaning day together with the lowest relative abundance of Firmicutes—compared to the AP supplemented piglets—may indicate towards a state of microbial dysbiosis, as also observed in young layer chicks [47].

It is worthy to note that the AP treatment seemed to affect more the composition of the microbiota of the mucosa than of the chyme and the composition of the microbiota in the 2% AP group did overall seem to differ more from the 0% and 4% AP groups.

Remarkably, no differences in SCFA between the different groups were observed in the feces on the 8 and 28 post-weaning days or in the caecum on the 35 post-weaning day. It is unclear why the differences in the diversity and composition of the microbiota seemed not to be reflected in changes in the SCFA.

### 4.4. AP as a Weaning Strategy for Gut Health

Taking together the data on performance, intestinal morphology and microbiota, the 4% AP diet showed potential to use as a feed ingredient for the post-weaning period in the experimental conditions applied, while the 2% AP diet seemed less interesting at the beginning of the post-weaning. It is uncertain which component of the AP was beneficial for the intestinal morphology and modulated the microbiota. AP used in the experiment was fairly rich in fructose, leading to more reducing sugars in the AP diets. As fructose may give energy to the small intestine [34] during the feed transition and was combined with an extra oil in AP diets, it may have affected the gut morphology. In this way, using a 4% AP diet during the post-weaning influenced positively the digestive process in the small intestine. In addition, the literature shows that the profile in dietary fiber of AP offers healthy components for the large intestine [1], amongst which pectin is known to exhibit positive effects on gut health parameters such as immunomodulation [48] or microbiota modulation [49], due to its impact on the digestive and fermentative processes [50]. However, the content of glucuronic acid and galacturonic acid in AP was rather low, which leaves the question if the pectins (<1 g/100 g DM) or the oligosaccharides (1 g/100 g DM of arabino-, xylo-, and galacto-oligosaccharides)—oligosaccharides that may also influence the immune response [51]—played a prebiotic effect [52,53,54]. Other important bioactive components of AP are the polyphenols; the AP used in the experiment seemed well endowed with it compared to cultivars quantified in the literature—twice [55] to five times more [11] than values found in the literature. In their review on polyphenols—gut microbiota and health-, Espin and colleagues [56] stated the emerging concept of 3P for gut health (probiotic, prebiotic and polyphenols) that promotes polyphenols to the same biological level of prebiotics. They explained the two-way interaction linking polyphenols and guts microbiota. Polyphenols shape the microbiota—enhancing the presence and abundance of bacteria beneficial for health—and microbiota catabolizes the polyphenols into metabolites often more active and absorbed by the colon than native forms. In a rat experiment on apple pectin and a polyphenol rich fraction of apple extract, Aprikian and colleagues [57] concluded that their combination is more effective than their separate supplementation. We support this hypothesis that AP exerts beneficial effects for health by its global matrix effect (involving fructose, oligosaccharides and other prebiotic components, and polyphenols) influencing in its entirety the digestive and the fermentative process in piglets.

The “gut health” principle consists of the equilibrium between the diet, the host (epithelium, mucus layer and gut-associated lymphoid tissue) and its microbiota (commensal bacteria and transient bacteria, including pathogens) [33]. The AP diets still need to be evaluated from a “host response” point of view to fully appreciate the effect of such a diet at weaning. Indeed, many mechanisms related to the barrier properties of the gastrointestinal tract are to be explored, such as permeability of the epithelium or interleukins and growth factors secretion [25]. From the bacterial equilibrium observed in the study, referring to the work of Spees and colleagues [44], a beneficial effect of the 4% AP diet can be expected on some immune factors. Moreover, Sehm and colleagues [9] showed a beneficial effect of AP on gut-associated lymphoid tissue by reducing its activity.

### 4.5. Emerging Concept from the Results

The nutrients in apple pomace may give a beneficial effect on the critical step of piglet’s weaning due to its monosaccharides, fiber, and polyphenols content by improving the energetic absorption—in the small intestine—and by modifying the gut microbiome.

At weaning, the digestive system of the piglet undergoes structural (intestinal architecture) and functional (enzymatic baggage) changes. The infant intestinal structure is replaced by a mature intestinal structure that is adapted to the new diet [15,24]. We hypothesized that during this transition, the monosaccharides profile of apple pomace maintained an energetic absorption—despite the global energetic deficiency status of piglet—by compensating the loss of the highly digestible carbohydrates of the milk.

Moreover, due to the weaning perturbations, the large intestine is potentially overloaded with readily fermentable nutrients—starch becoming a main fermentative substrate for the microbiota instead of the non-starch polysaccharides [58]. By incorporating apple pomace, the level of sugars in the diet progressively increased contrarily to the level of starch that decreased and apple pomace appeared beneficial in this way, enabling a good absorption of monosaccharides in the small intestine.

As a last identified consequence of adding apple pomace, the dietary fiber and polyphenolic compounds [2,3]—known to interact with the microbiota and to have beneficial health effect [56]—modulated the intestinal ecosystem so that the new balance resulted in healthy piglets, at least in this small study.

## 5. Conclusions

AP, at a level of 4% of incorporation, had positive effects on piglet’s performance, intestinal morphology and microbiota during the post-weaning period. A lower level of inclusion of AP—set at 2%—did not appear sufficient to induce these changes. AP constitutes a matrix of high interest for the feed sector due to its composition in dietary fiber (including oligosaccharides), biomolecules beneficial for health (as polyphenols), and probably also through its reduced sugars profile (fructose content). The results suggest that AP could be used as a suitable weaning strategy for gut health although the impact of the 4% AP diet on the gut epithelium and immune system need yet to be investigated. AP needs also to be investigated in large cohort to better evaluate the bacterial load of 4% AP diet in the context of the reduction of antibiotics in animal production and the phase-out of zinc oxide in weaner diets.

## Figures and Tables

**Figure 1 microorganisms-09-00572-f001:**
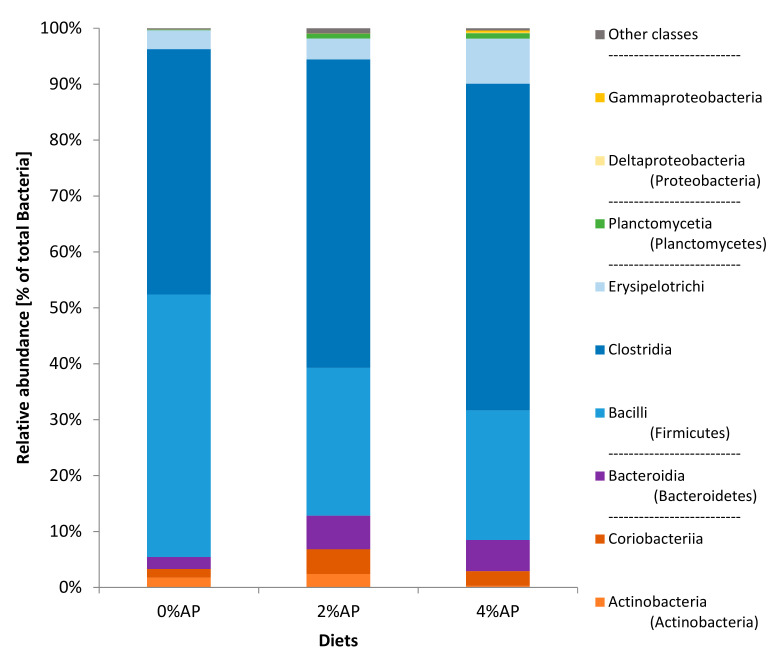
Composition (phyla and classes) of the microbiota in feces of piglets on the 8 post-weaning day. Diet 0% AP, control diet (*n* = 5), 2% AP, diet containing 2% dried apple pomace (*n* = 7), 4% AP, diet containing 4% dried apple pomace (*n* = 6).

**Figure 2 microorganisms-09-00572-f002:**
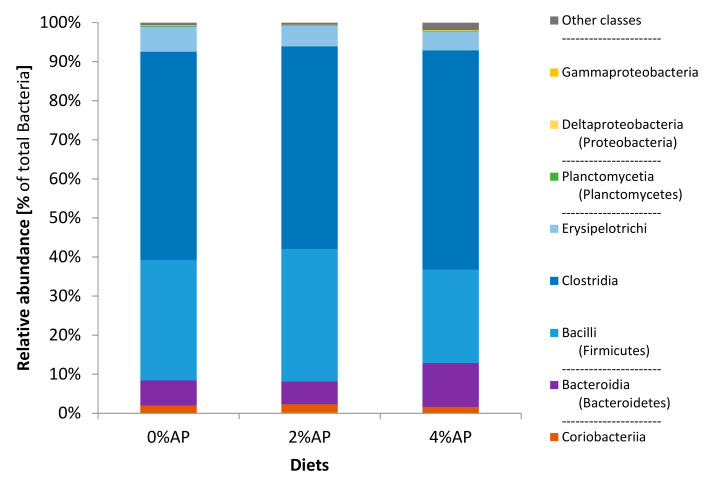
Composition (phyla and classes) of the microbiota in feces of piglets on the 28 post-weaning day. Diet 0% AP, control diet (*n* = 6), 2% AP, diet containing 2% dried apple pomace (*n* = 7), 4% AP, diet containing 4% dried apple pomace (*n* = 6).

**Table 1 microorganisms-09-00572-t001:** Composition of the post-weaning diets in %.

Ingredients	0% AP	2% AP	4% AP
Wheat	36.89	34.16	31.45
Barley	20.00	20.00	20.00
Soybean meal	16.40	16.83	17.28
Maize	15.00	15.00	15.00
Premix10916 (Inve Belgie, Dendermonde, Belgium)	7.50	7.50	7.50
Sugar beet pulp	2.50	2.50	2.50
AP	0.00	2.00	4.00
Soya oil	1.45	1.72	1.97
L-threonine	0.115	0.120	0.120
Monocalcium phosphate	0.100	0.100	0.100
Tryptophan	0.035	0.035	0.040
DL-Methionine	0.015	0.025	0.030
Rovimix^®^ E50	0.005	0.005	0.005

0% AP, control diet, 2% AP, 2% apple pomace diet, 4% AP, 4% apple pomace diet. Rovimix^®^ is a commercial source of vitamin E used in animal nutrition.

**Table 2 microorganisms-09-00572-t002:** Characterization of the apple pomace used in the study (mono-, and disaccharides, fiber and polyphenols).

Chemical Component	AP
Rhamnose	<0.1 g/100 g DM
Arabinose	<0.1 g/100 g DM
Xylose	<0.1 g/100 g DM
Mannose	<0.1 g/100 g DM
Glucose	3.4 g/100 g DM
Galactose	2.1 g/100 g DM
Fructose	13.5 g/100 g DM
Saccharose	2.9 g/100 g DM
Arabino-oligosaccharide	0.5 g/100 g DM
Xylo-oligosaccharide	0.3 g/100 g DM
Galacto-oligosaccharide	0.2 g/100 g DM
Galacturonic acid	<0.1 g/100 g DM
Glucuronic acid	<0.1 g/100 g DM
Total polyphenols (Folin-Ciocalteu)	26.7 mg gallic acid/g DM
Quercetin	23.1 µg/g DM
Phloretin	2.8 µg/g DM
Insoluble HMWDF	45.2 g/100 g DM
Soluble HMWDF	12.6 g/100 g DM
Soluble LMWDF	0.5 g/100 g DM
Soluble DF/insoluble DF	0.29

AP, apple pomace; DM, dry matter; HMWDF, high molar weight dietary fiber; LMWDF, low molar weight dietary fiber.

**Table 3 microorganisms-09-00572-t003:** Chemical analyses of the diets.

Chemical Component	0% AP	2% AP	4% AP
Dry Matter (%)	88.0	87.3	88.2
Crude protein (% DM)	18.1	18.7	18.6
Crude ash (% DM)	6.13	5.51	5.45
Reducing sugars (% DM)	1.43	1.76	2.47
Total sugars (% DM)	6.40	7.17	7.93
Starch (% DM)	44.10	43.41	42.23
NDF (% DM)	13.49	13.20	13.20
ADF (% DM)	6.49	6.56	6.68
ADL (% DM)	1.21	1.15	1.38
Cellulose (% DM)	5.13	5.12	4.93
Crude Fat (% DM)	3.34	3.55	3.50
Insoluble HMWDF (% DM)	17.3	17.4	16.7
Soluble HMWDF (% DM)	4.7	4.9	5.1
Soluble LMWDF (% DM)	3.3	3.0	3.1
Soluble DF/insoluble DF	0.46	0.45	0.49

ADF, acid detergent fiber; ADL, acid detergent lignin; DM, dry matter; HMWDF, high molar weight dietary fiber; LMWDF, low molar weight dietary fiber; NDF, neutral detergent fiber; 0% AP, control diet, 2% AP, 2% apple pomace diet, 4% AP, 4% apple pomace diet.

**Table 4 microorganisms-09-00572-t004:** Zootechnical performance of piglets receiving control or experimental diets during 35 days of the post-weaning rearing period.

Zootechnical Parameter	0% AP	2% AP	4% AP	SEM	*p*-Value Diet	*p*-Value Block
Initial weight (kg)	8.4	8.4	8.4	0.1	ns	ns
Final weight (kg)	21.0	22.6	23.1	0.4	ns	ns
Final ADG (kg/d)	0.361	0.406	0.421	0.012	ns	ns
TFI (kg DM)	19.8	21.7	20.5	0.5	ns	ns
FCR	1.79 ^a^	1.75 ^a^	1.59 ^b^	0.03	0.002	ns
EFE (MJ NE/kg gain)	16.9 ^a^	16.8 ^a^	15.2 ^b^	0.2	0.004	ns

^a, b^ values assigned different letter within a row are statistically different; 0% AP, control diet (*n* = 6); 2% AP, diet containing 2% dried apple pomace (*n* = 7); 4% AP, diet containing 4% dried apple pomace (*n* = 6); ADG, average daily gain; DM, dry matter; EFE, energetic feed efficiency; FCR, feed conversion ratio; NE, net energy; ns, not significant; TFI, total feed intake.

**Table 5 microorganisms-09-00572-t005:** Average daily weight gain (ADG) of piglets receiving control or experimental diets week by week.

ADG	0% AP	2% AP	4% AP	SEM	*p*-Value Diet	*p*-Value Block
ADG week 1 (d0–d + 7)	0.118	0.134	0.143	0.013	ns	ns
ADG week 2 (d + 7–d + 14)	0.304	0.261	0.350	0.020	0.083	ns
ADG week 3 (d + 14–d + 21)	0.367 ^b^	0.459 ^a^	0.458 ^a^	0.017	0.038	ns
ADG week 4 (d + 21–d + 28)	0.489	0.563	0.537	0.018	0.085	ns
ADG week 5 (d + 28–d + 35)	0.530	0.613	0.614	0.020	ns	ns

^a, b^ values assigned different letter within a row are statistically different; 0% AP, control diet (*n* = 6); 2% AP, diet containing 2% dried apple pomace (*n* = 7); 4% AP, diet containing 4% dried apple pomace (*n* = 6); ADG, average daily gain; d, day with d0 as the first day of the experiment.

**Table 6 microorganisms-09-00572-t006:** Contingency tables for the scoring of fecal consistency during the first 14 days post-weaning period and detection of the excreted pathogens in feces on the 8 post-weaning day.

Diet	Statistical Parameter	Fecal Consistency Scoring ^1^			Pathogens in Feces ^2^		
		A	B	Total	Negative	Positive	Total

Apple pomace	Count	51	144	195	84	20	104

	Expected count	57.47	137.53		87.58	16.42	
	Adjusted Residuals	−1.809	1.809		−1.713	1.713	
No apple pomace	Count	33	57	90	44	4	48

	Expected count	26.53	63.47		40.42	7.58	
	Adjusted Residuals	1.809	−1.809		1.713	−1.713	
Total	Count	84	201	285	128	24	152

*p*-value		0.096			0.072		

^1^ Contingency table from a Cochran test done on the maximum scores of fecal consistency. Feces were observed per box on a daily basis from the second day of weaning until the 15th post-weaning day. Consistency A included the results from no feces to normal feces (scores 0 to 3), consistency B included soft to liquid feces.^2^ Contingency table from a Chi-Square test of Association counting the presence (positive test or >10^6^ CFU) or absence (negative test or <10^6^ CFU) for Rotavirus, *E. Coli* F4, F5, F18, F41 attachment factors, *Clostridium difficile*, *Clostridium perfringens*, and Cryptosporidium. Fecal samples were taken on the 8 post-weaning day.

**Table 7 microorganisms-09-00572-t007:** Results of the morphological measurement (villus and crypts) on the three upper intestinal segments on the 35 post-weaning day.

Intestinal Parameter	0% AP	2% AP	4% AP	SEM	*p*-Value Diet	*p*-Value Block
Duodenum	(*n* = 6)	(*n* = 7)	(*n* = 6)			
Villus length (µm)	320 ^b^	381 ^ab^	429 ^a^	16.4	0.018	ns
Crypts depth (µm)	436	471	448	10.5	ns	ns
VL/CD	0.74	0.81	0.96	0.039	ns	ns
Jejunum	(*n* = 6)	(*n* = 6)	(*n* = 5)			
Villus length (µm)	354	366	420	20.5	ns	ns
Crypts depth (µm)	314	325	324	8.9	ns	ns
VL/CD	1.14	1.16	1.31	0.07	ns	ns
Ileum	(*n* = 6)	(*n* = 7)	(*n* = 6)			
Villus length (µm)	240	273	315	13.4	ns	ns
Crypts depth (µm)	255	270	242	7.9	ns	ns
VL/CD	0.94 ^b^	1.03 ^ab^	1.33 ^a^	0.07	0.039	ns

^a, b^ values assigned a different letter within a row are statistically different; 0% AP, control diet; 2% AP, diet containing 2% dried apple pomace; 4% AP, diet containing 4% dried apple pomace; VL/CD, ratio villus length/crypts depth.

**Table 8 microorganisms-09-00572-t008:** Relative abundances (% of total bacteria) of families and species of the microbiota in feces of piglets on the 28 post-weaning day.

Phylum-Class Family—Genus (If app.)—Species (If app.)	0%AP	2%AP	4%AP	SEM	*p*-Value KW/FDR
**Actinobacteria**–**Actinobacteria**-					
**Bifidobacteriaceae**	0.0	0.3	0.0	0.06	ns/ns
**Actinobacteria**–**Coriobacteriia**-					
**Coriobacteriaceae**	1.9	1.9	1.8	0.21	ns/ns
**Bacteroidetes**–**Bacteroidia**-					
Bacteroidales (undef. fam.)	0.1	0.2	0.3	0.06	ns/ns
**Prevotellaceae**	2.2	3.2	1.4	0.31	ns/ns
**Muribaculaceae** (formerly called S24-7)	4.0	2.5	8.7	1.46	ns/ns
(**Paraprevotellaceae**)	0.1	0.1	0.0	0.01	ns/ns
**Cyanobacteria**–4C0d-2					
YS2 (undef. fam.)	0.1	0.2	0.2	0.04	ns/ns
**Firmicutes**–**Bacilli**-					
**Enterococcaceae**	0.1	0.0	0.0	0.03	ns/ns
**Lactobacillaceae**	25.3	31.3	23.2	2.91	ns/ns
**Streptococcaceae**	6.1	3.0	1.8	0.84	ns/ns
**Firmicutes**–**Clostridia**-					
Clostridiales (Other fam.)	0.1	0.1	0.2	0.03	ns/ns
Clostridiales (undef. fam.)	3.7	2.7	5.5	0.44	0.099/ns
**Christensenellaceae**	1.5	0.9	3.1	0.61	ns/ns
**Clostridiaceae**	3.5 ^ab^	2.3 ^b^	5.9 ^a^	0.64	0.028/ns
Clostridiaceae -SMB53 (undef. sp)	0.6 ^ab^	0.4 ^b^	1.1 ^a^	0.14	0.036/ns
Clostridiaceae (undefined genus)	2.8 ^ab^	1.6 ^b^	4.6 ^a^	0.54	0.019/ns
**Dehalobacteriaceae**	0.1	0.0	0.1	0.02	ns/ns
**Eubacteriaceae**	0.0	0.2	0.0	0.05	ns/ns
**Lachnospiraceae**	16.3	19.8	14.1	1.50	ns/ns
Lachnospiraceae -*Blautia* (other sp.)	0.1 ^a^	0.0 ^b^	0.0 ^b^	0.02	0.024/ns
Lachnospiraceae -*Dorea* (other sp.)	0.8 ^a^	0.6 ^a^	0.2 ^b^	0.12	0.024/ns
Lachnospiraceae -*Lachnospira* (undef. sp)	0.4 ^ab^	0.4 ^a^	0.1 ^b^	0.06	0.036/ns
**Peptococcaceae**	0.2	0.1	0.2	0.02	ns/ns
**Peptostreptococcaceae**	0.1	0.1	0.1	0.02	ns/ns
**Ruminococcaceae**	13.6 ^ab^	12.9 ^b^	19.8 ^a^	1.27	0.047/ns
**Veillonellaceae**	12.2 ^a^	11.4 ^a^	4.8 ^b^	1.12	0.009/ns
Veillonellaceae -*Dialister* (undef. sp)	2.3 ^a^	2.1 ^ab^	0.5 ^b^	0.32	0.016/ns
Veillonellaceae -*Megasphaera* (undef. sp)	5.7 ^a^	6.6 ^a^	2.8 ^b^	0.59	0.015/ns
Veillonellaceae -*Mitsuokella* (undef. sp)	2.4 ^a^	1.1 ^b^	0.6 ^b^	0.29	0.023/ns
Veillonellaceae -*Mitsuokella multacida*	0.1	0.5	0.1	0.09	0.063/ns
Veillonellaceae -(undef. genus)	0.5	0.3	0.1	0.08	0.064/ns
[**Mogibacteriaceae**]	1.4	1.2	1.6	0.16	ns/ns
**Firmicutes**–**Erysipelotrichi**-					
**Erysipelotrichaceae**	6.6	5.0	5.3	0.62	ns/ns
Erysipelotrichaceae -[*Eubacterium*] *cylindroides*	0.7	0.3	0.6	0.09	0.092/ns
Erysipelotrichaceae -*Bulleidia* p-1630-c5	1.3	0.9	0.6	0.01	0.085/ns
**Erysipelotrichaceae -L7A_E11 (undef. sp)**	0.0	0.0	0.1	0.03	0.060/ns
Erysipelotrichaceae -(undef. genus)	0.1	0.1	0.2	0.02	0.085/ns
**Planctomycetes**–**Planctomycetia**-					
**Pirellulaceae**	0.2	0.2	0.2	0.08	ns/ns
**Proteobacteria**–**Deltaproteobacteria**-					
**Desulfovibrionaceae**	0.1	0.0	0.1	0.02	0.071/ns
-*Desulfovibrio* (undef. sp)	0.1	0.0	0.1		0.100/ns
**Proteobacteria**–**Gammaproteobacteria**-					
**Succinivibrionaceae**	0.0	0.1	0.0	0.01	ns/ns
TM7–TM7-3-					
F16	0.0	0.1	0.1	0.03	ns/ns
**Tenericutes**–**Mollicutes**-					
RF39 (undef. fam.)	0.1	0.1	0.1	0.01	ns/ns
WPS-2—(undefined class)-					
(undef. fam.)	0.0	0.0	0.6	0.18	0.065/ns

Families showing a relative abundance > 0.1% in at least one diet were included. Only species showing a statistical trend or significance were included; relative abundance in % of the total bacteria. ^a, b^ values assigned a different letter within a row are statistically different; 0%AP, control diet (*n* = 6); 2%AP, diet containing 2% dried apple pomace (*n* = 7); 4%AP, diet containing 4% dried apple pomace (*n* = 6); if app., if applicable; KW/FDR, Kruskal–Wallis/Benjamini–Hochberg false discovery rate; ns = not significant; undef. fam., undefined family; undef. sp, undefined species.

**Table 9 microorganisms-09-00572-t009:** Alpha-diversity indexes of the microbiota from the chyme and the mucosa of the caecum on the 35 post-weaning day.

Index	0% AP	2% AP	4% AP	*p*-Value KW
Chyme				
Chao 1	1103 ^ab^	990 ^b^	1189 ^a^	0.015
OTU	707 ^ab^	633 ^b^	790 ^a^	0.024
PD Whole Tree	40.0 ^ab^	36.1 ^b^	44.3 ^a^	0.030
Shannon	5.4	5.3	5.7	ns
Mucosa				
Chao 1	771 ^ab^	668 ^b^	849 ^a^	0.045
OTU	491	423	527	0.084
PD Whole Tree	34.1	30.2	36.4	ns
Shannon	6.0	5.5	6.1	ns

^a, b^ values assigned a different letter within a row are statistically different; 0% AP, control diet; 2% AP, diet containing 2% dried apple pomace; 4% AP, diet containing 4% dried apple pomace; KW, Kruskal–Wallis; ns, not significant; OTU, operational taxonomic unit; PD, phylogenetic diversity.

**Table 10 microorganisms-09-00572-t010:** Relative abundances (% of total bacteria) of the microbiota from the mucosa of the caecum in piglets on the 35 post-weaning day.

Phylum-Class: Family–Genus–Species	0% AP	2% AP	4% AP	SEM	*p*-Value KW/FDR
Bacteroidetes	14.6 ^a^	6.5 ^b^	13.6 ^a^	1.27	0.049/ns
Bacteroidia:	14.6 ^a^	6.5 ^b^	13.6 ^a^	1.27	0.049/ns
Prevotellaceae	11.9	5.2	10.5	1.08	0.060/ns
Prevotellaceae–*Prevotella* (undef. sp.)	3.0 ^a^	1.0 ^b^	3.6 ^a^	0.36	0.010/ns
Muribaculaceae (formerly called s24-7)	0.2 ^ab^	0.1 ^b^	0.3 ^a^	0.03	0.037/ns
Firmicutes	78.3 ^b^	90.1 ^a^	82.4 ^b^	1.68	0.045/ns
Clostridia:	46.5	49.4	54.5	2.02	ns/ns
Lachnospiraceae–*Coprococcus* (undef. sp.)	0.7	0.5	0.9	0.07	0.073/ns
Peptostreptococcaceae	0.1	0.1	0.1	0.01	0.074/ns
Ruminococcaceae–*Oscillospira* (undef. sp.)	0.2 ^a^	0.1 ^b^	0.3 ^a^	0.03	0.037/ns
Veillonellaceae–*Acidaminococcus* (undef. sp.)	0.1	0.2	0.3	0.04	0.088/ns
Veillonellaceae–*Megamonas* (undef. sp.)	0.4	3.5	0.1	0.97	0.070/ns
Veillonellaceae–*Mitsuokella multacida*	0.0	0.5	0.1	0.10	0.094/ns
Proteobacteria	5.6	1.3	3.0	0.63	0.053/ns
Epsilonproteobacteria:	3.2	0.9	1.4	0.39	ns/ns
Campylobacteraceae–*Campylobacter* (undef. sp.)	2.5	0.5	0.9	0.34	0.083/ns
Gammaproteobacteria:	2.1	0.3	1.4	0.33	ns/ns
Pasteurellaceae	0.2	0.0	0.0	0.03	0.057/ns
Pasteurellaceae–*Actinobacillus* (Other sp.)	0.1 ^a^	0.0 ^b^	0.0 ^a^	0.02	0.030/ns

^a, b^ values assigned a different letter within a row are statistically different; 0% AP, control diet (*n* = 6); 2% AP, diet containing 2% dried apple pomace (*n* = 7); 4% AP, diet containing 4% dried apple pomace (*n* = 6); KW/FDR, Kruskal–Wallis/Benjamini–Hochberg false discovery rate; ns, not significant; OTU, operational taxonomic unit; PD, phylogenetic diversity; undef. sp., undefined species. Are shown in the table the bacteria with statistical differences.

## Data Availability

The datasets analyzed for this study can be found in the ENA database (https://www.ebi.ac.uk/ena/browser/home, accessed on 30 June 2020) under the accession number PRJEB38962.

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
