# Peer review of "Apple Pomace and Performance, Intestinal Morphology and Microbiota of Weaned Piglets—A Weaning Strategy for Gut Health?"

_microorganisms, 2021, doi:10.3390/microorganisms9030572_

Round 1

Reviewer 1 Report

Dufourny et al. described an interesting study by using 4% apple pomace (AP) as feed ingredient for weaning piglets, and showed positive effects on their performance, intestinal morphology and microbiota during the post-weaning period. The research design is sound, the methodologies applied are adequate and carefully described, the results are clearly presented, and the conclusions are well supported by the results. In my opinion, this manuscript is suitable for publication in the journal of Microorganisms, however, there are still some minor points that need to be addressed as follows, before it can be accepted for publication.

  1. In Figures 2 and 3, the authors used bar charts to describe the composition (phyla and classes) of the microbiota in faeces of piglets on the 8th and 28th post-weaning day, respectively. To clearly present the differentially abundant genera of each experimental group, "Species abundance heat map" should also be provided.
  2. In Appendix A, the authors used "Principal component analysis plots" (Weighted Unifrac distance metric) for the faecal samples and chyme samples and mucosal samples from caecum. To better describe the results and clearly present the differences between each group, "Beta diversity analysis plots" should also be provided.  

Author Response

Dear Reviewer 1,

For the comment 1: Two Heat Maps were created (one for the faeces on 8 post-weaning day and one for the faeces on 28 post-weaning day). The results did not help in visualizing the differences between treatments and appeared not conclusive. So we preferred to add the maps in supplementary file (with appropriate reference in the manuscript). 

For the comment 2 : we only have at our disposal - as beta diversity plots - the “Principal Coordinate Analysis of beta diversity”. We updated the title of the appendix A to clarify this point.

Yours faithfully

Reviewer 2 Report

This is a clearly written manuscript to which I have the following suggestions for minor improvement.

l.20, do not start sentence with number

l.21, what is PWday? Avoid laboratory slang and abbreviations

l.56, delete “methods“

l.91, „; 4%AP being similar to 0%AP“. Please, reword. This is not a sentence, verb is missing.

  1. 128, why only males? Please explain

l.146, replace „following their internal protocols. They concern“ with „using“

l.147, delete „method - previously described“. Then delete also the second sign „-„

l.148, 149, delete „method – previously described“

l.155, specify the solvent

  1. 185 and throughout the whole manuscript, correct two points at the end of several numerical bullets

l.193, revert order, 105 formol

l.239, ... and Shannon index.

l.271, 106, use upper index in both cases at this line

I do not insist but I recommend to delete either table 5 or fig. 1 since these show the same data

l.350, replace „realized“ with „performed“

l.371, „3.6.1. On the 8th PWday in the faecal samples“ is low quality expression. Say Faecal microbiota composition on day 8 post weaning. The same style of corrections also applies for line 404, 446, 447 and 463

l.379, delete „It was followed“ and reword accordingly. The same stands also for l.420

l.452, delete „rather“

l.469, delete sentence „It was observed for the two dominant phyla.“

l.472, use singular, phylum

l.474 and 475, delete „It concerned“ and reword. Use simple nice sentences.

l.522, delete sentence „No antibiotics were used.“

l.580, delete „interesting“

l.583, delete „our“

l.589 and 590, delete (2020) and leave only names of authors followed by regular reference according to style of Microorganisms

l.591, „wrote about“ is little bit naive, please reword

l.599, do not split polyphenols into two lines

l.603, correct multiplicated reference

Author Response

Dear Reviewer 2,

Yours faithfully
